# Radial Basis Function Finite Difference Method Based on Oseen Iteration for Solving Two-Dimensional Navier–Stokes Equations

**DOI:** 10.3390/e25050804

**Published:** 2023-05-16

**Authors:** Liru Mu, Xinlong Feng

**Affiliations:** College of Mathematics and System Sciences, Xinjiang University, Urumqi 830017, China; muliru2021@163.com

**Keywords:** Navier–Stokes equation, radial basis function finite difference method, polynomial, Oseen iteration

## Abstract

In this paper, the radial basis function finite difference method is used to solve two-dimensional steady incompressible Navier–Stokes equations. First, the radial basis function finite difference method with polynomial is used to discretize the spatial operator. Then, the Oseen iterative scheme is used to deal with the nonlinear term, constructing the discrete scheme for Navier–Stokes equation based on the finite difference method of the radial basis function. This method does not require complete matrix reorganization in each nonlinear iteration, which simplifies the calculation process and obtains high-precision numerical solutions. Finally, several numerical examples are obtained to verify the convergence and effectiveness of the radial basis function finite difference method based on Oseen Iteration.

## 1. Introduction

The Navier–Stokes equations are a set of equations used to describe fluid substance such as liquid and air, and can be used to simulate weather, ocean currents, water flows in pipes, the motion of stars in galaxies, etc. Therefore, they are of significant research value; this paper discusses the numerical solution method of steady incompressible Navier–Stokes equations.

Scholars have studied many numerical methods for Navier–Stokes equations [1]. For unsteady Navier–Stokes problems [2] and steady Navier–Stokes problems [3], a variety of traditional numerical methods have been proposed, including the finite element method, finite difference method, and finite volume method. In response to incompressible conditions, nonlinearity, long-time integration, and other difficulties encountered in numerical solutions of three-dimensional unsteady Navier–Stokes equations, this paper discusses the research status and latest research results of highly efficient and fully discrete finite element methods, which can be used to overcome these difficulties. In addition, this paper illustrates stability and error estimation of finite element-space discrete solutions and optimal error estimation of efficient fully-discrete finite element solutions for solving three-dimensional unsteady Navier–Stokes equations [4]. These methods are mesh-based, and the generation of meshes increases computational cost; moreover, the computational domain and the technology used for mesh quality and stabilization affect the computational accuracy. Especially for high-dimensional problems, it is difficult and costly to generate good meshes. Therefore, in recent years many scholars have introduced meshless methods [5,6] for solving partial differential equations, among which the radial basis function method [7] is increasingly popular.

The radial basis function method has been widely used for solving various partial differential equations, with the radial basis function finite difference method (RBF-FD) proposed considering ill-conditioned problems with a high occurrence rate in the matrix generated by the global radial basis function method. In this way, it is easy to realize RBF-FD discretization without needing to use a mesh at all. The RBF-FD method can be regarded as an extension of the standard finite difference and local refinement of global radial basis function method. This method uses the radial basis function to create weights for the RBF-FD formula, and the differential matrix obtained in this way is as sparse as that obtained by the standard difference method, which combines many advantages of radial basis function and traditional finite difference approximation. Thus, RBF-FD approximation [8] is an attractive substitute for the global radial basis function method.

In order to control the stagnation error, Natasha Flyer et al. introduced higher-order polynomials under the background of Gaussian (GA) and Polyharmonic Spline (PHS) interpolation [9,10]. PHS eliminates the stagnation error in combination with the polynomial to capture the basic physical properties of the problem, and achieves high-order accuracy without the need to adjust the shape parameters. The present paper chooses the PHS basis function and applies the radial basis function finite difference method with the polynomial to discretize the spatial operator of the equation. It is well known that the iterative method [11] is effective for Navier–Stokes equations with strong nonlinear terms, e.g., the Newton iterative method, Stokes iterative method, and Oseen iterative method. This paper focuses on the Oseen iterative method to deal with the nonlinear terms of the equation. The radial basis function finite difference method based on Oseen iteration proposed in this paper does not need a complete set of matrices in each nonlinear iteration, which is more suitable for nonlinear problems, and obtains high-precision numerical solutions.

## 2. Problem Setup

This paper studies the following two-dimensional steady incompressible Navier–Stokes equations [12].
(1)$$\begin{array}{c} \begin{array}{cc} -\nu \Delta u+(u\cdot \nabla )u+\nabla p & =f\;\;\;\mathrm{on}\;\Omega ,\\ \nabla \cdot u & =0\;\;\;\mathrm{on}\;\Omega ,\\ u & =0\;\;\;\mathrm{on}\;\partial \Omega . \end{array} \end{array}$$
where Ω is the bounded region on $R^{2}$, $u=\left(u,v\right):\Omega \to R^{2}$ is the velocity vector, p:Ω→R is the pressure, ν>0 is the viscosity coefficient, and $f=(f_{1},f_{2})$ is the external force acting on a unit volume of fluid, assuming that the uniqueness of *p* can be achieved by imposing the condition, $\int_{\Omega }p\;dx=0$.

We define the 2-norm and *∞*-norm of the vector x as
$${∥x∥}_{2}={\left(\sum_{i=1}^{n}x_{i}^{2}\right)}^{\frac{1}{2}}{,\;\;\;\;\;∥x∥}_{\infty }=\underset{1\leq i\leq n}{max}\left|x_{i}\right|.$$

## 3. Numerical Method

In this part, we focus on the radial basis function finite difference method with polynomial.

### 3.1. Radial Basis Function Finite Difference Method with Polynomial

First, this part provides a brief introduction to the radial basis function (RBF). The RBF, expressed as $\Phi \left(r\right):R^{d}\to R$, is a d-dimensional radially symmetric function, and is only dependent on $r=∥x-x_{k}{∥}_{2}$, where ${∥\cdot ∥}_{2}$ represents the Euclidean distance between two points, x is the point to be solved, and $x_{k}$ is the center position of the RBF. Common radial basis functions include Gauss (GA) ($e^{-{\left(\varepsilon r\right)}^{2}}$), multiquadratic function ($\sqrt{1+{\left(\varepsilon r\right)}^{2}}$), and Polyharmonic Spline (PHS) ($r^{m},m=1,3,5,\cdots$), where ε is the shape parameter that determines the radial basis function. Next, we introduce the radial basis function finite difference method with polynomial. First, *L* is defined as a linear operator, which can be Δ, $\frac{\partial }{\partial x}$, $\frac{\partial }{\partial y}$, etc. We can approximate the value of the operator *L* at node $x_{c}$ by linear combination of the function value ${\left\{u_{k}\right\}}_{k=1}^{n}$. For this, we need to select *n* nodes near the central node $x_{c}$ to form a node template, which is recorded as $X={\left\{x_{k}\right\}}_{k=1}^{n}$, including $x_{c}$, as follows:(2)$$\begin{array}{c} \begin{array}{c} \left(Lu\right){|}_{x=x_{c}}=\sum_{k=1}^{n}w_{k}u_{k}, \end{array} \end{array}$$
where *n* is the size of the node template and $w_{k}$ is the differential weight. In order to calculate the weight $w_{k}$, we can specify that the linear combination of function values be accurate for interpolation s(x):(3)$$\begin{array}{c} \begin{array}{c} s\left(x\right)=\sum_{k=1}^{n}\lambda_{k}\Phi \left(∥x-x_{k}{∥}_{2}\right)+\sum_{k=1}^{\ell }\mu_{k}p_{k}\left(x\right), \end{array} \end{array}$$
The constraint condition is
(4)$$\begin{array}{c} \begin{array}{c} \sum_{k=1}^{n}\lambda_{k}p_{j}\left(x_{k}\right)=0\;\;j=1,2\cdots \ell , \end{array} \end{array}$$
where $p_{l}\left(x\right)$ is degree *l* of the binary polynomial and $\Phi \left(∥x-x_{k}{∥}_{2}\right)$ is the radial basis function.

When d=2 and the degree of the polynomial is l=1, Equation (Equation 3) is
(5)$$\begin{array}{c} \begin{array}{c} s\left(x\right)=\sum_{k=1}^{n}\lambda_{k}\Phi \left(∥x-x_{k}{∥}_{2}\right)+\mu_{1}+\mu_{2}x+\mu_{3}y, \end{array} \end{array}$$
with the constraint condition (4)
(6)$$\begin{array}{c} \begin{array}{c} \sum_{k=1}^{n}\lambda_{k}=\sum_{k=1}^{n}\lambda_{k}x_{k}=\sum_{k=1}^{n}\lambda_{k}y_{k}=0. \end{array} \end{array}$$
Suppose that the function value at $x_{k}$ is $f_{k},k=1,\cdots ,n$; then, the matrix form of Equations (5) and (6) is
$$\left[\begin{array}{cccccc} \Phi (∥x_{1}-x_{1}{∥}_{2}) & \cdots & \Phi (∥x_{1}-x_{n}{∥}_{2}) & 1 & x_{1} & y_{1}\\ \Phi (∥x_{2}-x_{1}{∥}_{2}) & \cdots & \Phi (∥x_{2}-x_{n}{∥}_{2}) & 1 & x_{2} & y_{2}\\ \vdots & \vdots & \vdots & \vdots & \vdots & \vdots \\ \Phi (∥x_{n}-x_{1}{∥}_{2}) & \cdots & \Phi (∥x_{n}-x_{n}{∥}_{2}) & 1 & x_{n} & y_{n}\\ 1 & \cdots & 1 & 0 & 0 & 0\\ x_{1} & \cdots & x_{n} & 0 & 0 & 0\\ y_{1} & \cdots & y_{n} & 0 & 0 & 0 \end{array}\right]\left[\begin{array}{c} \lambda_{1}\\ \lambda_{2}\\ \vdots \\ \lambda_{n}\\ \mu_{1}\\ \mu_{2}\\ \mu_{3} \end{array}\right]=\left[\begin{array}{c} f_{1}\\ f_{2}\\ \vdots \\ f_{n}\\ 0\\ 0\\ 0 \end{array}\right],$$
In this formula, $\hat{A}$ is used to represent the matrix at the left end (n+3)×(n+3); thus,
(7)$$\begin{array}{c} \begin{array}{c} s\left(x\right)=[\Phi (∥x-x_{1}{∥}_{2})\ldots \Phi \left(∥x-x_{n}{∥}_{2}\right)\;1\;x\;y]{\left[\lambda_{1}\ldots \lambda_{n}\;\mu_{1}\;\mu_{2}\;\mu_{3}\right]}^{T}\\ =[\Phi (∥x-x_{1}{∥}_{2})\ldots \Phi \left(∥x-x_{n}{∥}_{2}\right)\;1\;x\;y]{\hat{A}}^{-1}{\left[f_{1}\ldots f_{n}\;0\;0\;0\right]}^{T}, \end{array} \end{array}$$
The linear operator *L* is used to calculate the result at $x=x_{c}$:(8)$$\begin{array}{c} \begin{array}{c} {Ls\left(x\right)|}_{x=x_{c}}=[L\Phi (∥x-x_{1}{∥}_{2}{)|}_{x=x_{c}}\ldots L\Phi \left(∥x-x_{n}{∥}_{2}\right){|}_{x=x_{c}}\;\\ {L1|}_{x=x_{c}}{\;Lx|}_{x=x_{c}}\;Ly{|}_{x=x_{c}}]{\hat{A}}^{-1}{\left[f_{1}\ldots f_{n}\;0\;0\;0\right]}^{T}. \end{array} \end{array}$$
If $f_{i}=1$ and $f_{j}=0$, then j≠i; thus, ${Ls\left(x\right)|}_{x=x_{c}}=w_{i}$ and i=1,⋯,n, according to Formula (2). Therefore,
$$\left[w_{1},\cdots ,w_{n}\right]=[L\Phi (∥x-x_{1}{∥}_{2}{)|}_{x=x_{c}}\ldots L\Phi \left(∥x-x_{n}{∥}_{2}\right){|}_{x=x_{c}}\;$$
$${L1|}_{x=x_{c}}{\;Lx|}_{x=x_{c}}\;Ly{|}_{x=x_{c}}]{\hat{A}}^{-1}{\left[\begin{array}{c} D\\ O \end{array}\right]}_{(n+3)\times n},$$
where *D* is the identity matrix of (n×n) and *O* is the null matrix of (3×n). An identity matrix is formed by adding three columns to the right matrix, which can be ignored. Thus, the left-hand matrix is added with $\left[w_{n+1},w_{n+2},w_{n+3}\right]$. The right-hand matrix is multiplied by $\hat{A}$ and then transposed, obtaining the following linear equations:$$\left[\begin{array}{cccccc} \Phi (∥x_{1}-x_{1}{∥}_{2}) & \cdots & \Phi (∥x_{1}-x_{n}{∥}_{2}) & 1 & x_{1} & y_{1}\\ \Phi (∥x_{2}-x_{1}{∥}_{2}) & \cdots & \Phi (∥x_{2}-x_{n}{∥}_{2}) & 1 & x_{2} & y_{2}\\ \vdots & \vdots & \vdots & \vdots & \vdots & \vdots \\ \Phi (∥x_{n}-x_{1}{∥}_{2}) & \cdots & \Phi (∥x_{n}-x_{n}{∥}_{2}) & 1 & x_{n} & y_{n}\\ 1 & \cdots & 1 & 0 & 0 & 0\\ x_{1} & \cdots & x_{n} & 0 & 0 & 0\\ y_{1} & \cdots & y_{n} & 0 & 0 & 0 \end{array}\right]\left[\begin{array}{c} w_{1}\\ w_{2}\\ \vdots \\ w_{n}\\ w_{n+1}\\ w_{n+2}\\ w_{n+3} \end{array}\right]=\left[\begin{array}{c} L\Phi (∥x-x_{1}{∥}_{2}){|}_{x=x_{c}}\\ L\Phi (∥x-x_{2}{∥}_{2}){|}_{x=x_{c}}\\ \vdots \\ L\Phi (∥x-x_{n}{∥}_{2}){|}_{x=x_{c}}\\ L1{|}_{x=x_{c}}\\ Lx{|}_{x=x_{c}}\\ Ly{|}_{x=x_{c}} \end{array}\right].$$
Therefore, the weights $w_{1}\cdots w_{n}$ are obtained from the above equations. In the following numerical examples, the resulting linear system changes with the varying degree of the polynomial. In this paper, the PHS radial basis function is mainly considered; $\Phi \left(r\right)=r^{m},m=1,3,5,\cdots$, PHS is a piecewise smooth function, where $r=∥x-x_{k}{∥}_{2}$, and it is unnecessary to select a shape parameter ε.

### 3.2. Radial Basis Function Finite Difference Method Based on Oseen Iteration

In this part, we introduce discretization of the Navier–Stokes equations using the radial basis function finite difference method based on Oseen iteration. First, we use the radial basis function finite difference method with polynomial to discretize the component equations. Then, we linearize the equations based on Oseen iteration starting from this scheme, and finally obtain the fully discrete scheme of the equations.

#### 3.2.1. Discrete Scheme of Equation

First, the radial basis function finite difference method with polynomial is used to discretize the spatial operator of the equation. Equation (Equation 1) is expressed as the following components:(9)$$\begin{array}{c} \begin{array}{cc} -\nu \left(\frac{\partial^{2}u}{\partial x^{2}}+\frac{\partial^{2}u}{\partial y^{2}}\right)+u\frac{\partial u}{\partial x}+v\frac{\partial u}{\partial y}+\frac{\partial p}{\partial x} & =f_{1}, \end{array} \end{array}$$
(10)$$\begin{array}{c} \begin{array}{cc} -\nu \left(\frac{\partial^{2}v}{\partial x^{2}}+\frac{\partial^{2}v}{\partial y^{2}}\right)+u\frac{\partial v}{\partial x}+v\frac{\partial v}{\partial y}+\frac{\partial p}{\partial y} & =f_{2}, \end{array} \end{array}$$
(11)$$\begin{array}{c} \begin{array}{cc} \frac{\partial u}{\partial x}+\frac{\partial v}{\partial y} & =0. \end{array} \end{array}$$
For Equation (Equation 9), the differential value at a point $x_{c}$ is
(12)$$\begin{array}{c} \begin{array}{c} Lu\left(x_{c}\right)=\sum_{j=1}^{n}w_{j}u\left(x_{cj}\right), \end{array} \end{array}$$
where ${\left\{x_{cj}\right\}}_{j}^{n}$ is the *n* points around $x_{c}$ and ${\left\{w_{j}\right\}}_{j}^{n}$ is the weight coefficient corresponding to these points obtained by applying the interpolation condition $f\left(x_{cj}\right)=f_{j}$.

First, we assume that there are N nodes in a zone. At a certain point $x_{c}$, Formula (9) is discretized according to RBF-FD interpolation and the following form is obtained:(13)$$\begin{array}{c} \begin{array}{c} -\nu \sum_{j=1}^{n_{1}}w_{j}^{\Delta }u\left(x_{cj}\right)+u\left(x_{c}\right)\sum_{j=1}^{n_{1}}w_{j}^{\nabla_{x}}u\left(x_{cj}\right)+v\left(x_{c}\right)\sum_{j=1}^{n_{1}}w_{j}^{\nabla_{y}}u\left(x_{cj}\right)+\sum_{j=1}^{n_{2}}w_{j}^{\nabla_{x}}p\left(x_{cj}\right)\\ =f_{1}\left(x_{c}\right), \end{array} \end{array}$$
where $n_{1}$ represents the number of nearest nodes at the central node $x_{c}$, used for approximating the weight coefficients of Δu and ∇u, and $n_{2}$ represents the number of nodes at the central node $x_{c}$, used to form a template of nearest nodes for approximating the weight coefficients of ∇p; moreover, $w_{j}^{\Delta }$ is a weight coefficient for approximating Δu, $w_{j}^{\nabla_{x}}$ is a weight coefficient for approximating operator $\frac{\partial }{\partial x}$, and $w_{j}^{\nabla_{y}}$ is a weight coefficient for approximating operator $\frac{\partial }{\partial y}$. Next, we use the Oseen iterative scheme to deal with the nonlinear terms of the steady Navier–Stokes equation. Iterating according to Discrete Equation (Equation 13), we obtain the following form:(14)$$\begin{array}{c} \begin{array}{c} -\nu \sum_{j=1}^{n_{1}}w_{j}^{\Delta }u\left(x_{cj}^{k+1}\right)+u\left(x_{c}^{k}\right)\sum_{j=1}^{n_{1}}w_{j}^{\nabla_{x}}u\left(x_{cj}^{k+1}\right)+v\left(x_{c}^{k}\right)\sum_{j=1}^{n_{1}}w_{j}^{\nabla_{y}}u\left(x_{cj}^{k+1}\right)+\sum_{j=1}^{n_{2}}w_{j}^{\nabla_{x}}p\left(x_{cj}^{k+1}\right)\\ =f_{1}\left(x_{c}^{k+1}\right), \end{array} \end{array}$$
then
(15)$$\begin{array}{c} \begin{array}{c} \left(-\nu \sum_{j=1}^{n_{1}}w_{j}^{\Delta }+u\left(x_{c}^{k}\right)\sum_{j=1}^{n_{1}}w_{j}^{\nabla_{x}}+v\left(x_{c}^{k}\right)\sum_{j=1}^{n_{1}}w_{j}^{\nabla_{y}}\right)u\left(x_{cj}^{k+1}\right)+\sum_{j=1}^{n_{2}}w_{j}^{\nabla_{x}}p\left(x_{cj}^{k+1}\right)=f_{1}\left(x_{c}^{k+1}\right), \end{array} \end{array}$$
where *k* denotes the number of iterations, $u(x_{c}^{k})$ denotes the value of *u* at the *k*th iteration, $u(x_{c}^{k+1})$ denotes the value of *u* at the (*k* + 1)th iteration, and the initial value of the velocity $u^{0}$ is obtained from the corresponding Stokes equation. In Equation (Equation 15), the value of *u* at the *k*th level is known; thus, the nonlinear term in Equation (Equation 9) is linearized. Likewise, there are similar discrete forms for Equations (10) and (11):(16)$$\begin{array}{c} \begin{array}{c} \left(-\nu \sum_{j=1}^{n_{1}}w_{j}^{\Delta }+u\left(x_{c}^{k}\right)\sum_{j=1}^{n_{1}}w_{j}^{\nabla_{x}}+v\left(x_{c}^{k}\right)\sum_{j=1}^{n_{1}}w_{j}^{\nabla_{y}}\right)v\left(x_{cj}^{k+1}\right)+\sum_{j=1}^{n_{2}}w_{j}^{\nabla_{y}}p\left(x_{cj}^{k+1}\right)=f_{2}\left(x_{c}^{k+1}\right), \end{array} \end{array}$$
(17)$$\begin{array}{c} \begin{array}{c} \sum_{j=1}^{n_{1}}w_{j}^{\nabla_{x}}u\left(x_{cj}^{k+1}\right)+\sum_{j=1}^{n_{1}}w_{j}^{\nabla_{y}}v\left(x_{cj}^{k+1}\right)=0, \end{array} \end{array}$$
There are *N* nodes in a zone including $N_{B}$ boundary nodes and $N_{I}$ internal nodes, where $N=N_{I}+N_{B}$. The central node $x_{c}$ in Formulas (15) and (16) is considered on $N_{I}$ internal nodes, and the central node $x_{c}$ in Formula (17) is considered on N total nodes. Considering the boundary condition u=0, Equations (15)–(17) and the boundary conditions form the following equations:(18)$$\begin{array}{c} \begin{array}{c} \left[\begin{array}{ccccc} W_{N_{I}\times N_{I}}^{\Delta +\nabla } & W_{N_{I}\times N_{B}}^{\Delta +\nabla } & O_{N_{I}\times N_{I}} & O_{N_{I}\times N_{B}} & W_{N_{I}\times N}^{\nabla_{x}}\\ O_{N_{B}\times N_{I}} & E_{N_{B}\times N_{B}} & O_{N_{B}\times N_{I}} & O_{N_{B}\times N_{B}} & O_{N_{B}\times N}\\ O_{N_{I}\times N_{I}} & O_{N_{I}\times N_{B}} & W_{N_{I}\times N_{I}}^{\Delta +\nabla } & W_{N_{I}\times N_{B}}^{\Delta +\nabla } & W_{N_{I}\times N}^{\nabla_{y}}\\ O_{N_{B}\times N_{I}} & O_{N_{B}\times N_{B}} & O_{N_{B}\times N_{I}} & E_{N_{B}\times N_{B}} & O_{N_{B}\times N}\\ W_{N\times N_{I}}^{\nabla_{x}} & W_{N\times N_{B}}^{\nabla_{x}} & W_{N\times N_{I}}^{\nabla_{y}} & W_{N\times N_{B}}^{\nabla_{y}} & O_{N\times N} \end{array}\right]\left[\begin{array}{c} U_{I}\\ U_{B}\\ V_{I}\\ V_{B}\\ P \end{array}\right]=\left[\begin{array}{c} F_{1}\\ 0\\ F_{2}\\ 0\\ 0\; \end{array}\right], \end{array} \end{array}$$
We can now write (18) in the form of a block matrix:(19)$$\begin{array}{c} \begin{array}{c} \left[\begin{array}{ccc} W_{u}^{\Delta +\nabla } & O & W_{p}^{\nabla_{x}}\\ O & W_{v}^{\Delta +\nabla } & W_{p}^{\nabla_{y}}\\ W_{u}^{\nabla_{x}} & W_{v}^{\nabla_{y}} & O \end{array}\right]\left[\begin{array}{c} U\\ V\\ P \end{array}\right]=\left[\begin{array}{c} F_{1}\\ F_{2}\\ 0\; \end{array}\right]. \end{array} \end{array}$$The stabilization parameter ϵ is added to Equation (Equation 19) to obtain a system of equations in the form of BU = F:(20)$$\begin{array}{c} \begin{array}{c} \left[\begin{array}{ccc} W_{u}^{\Delta +\nabla } & O & W_{p}^{\nabla_{x}}\\ O & W_{v}^{\Delta +\nabla } & W_{p}^{\nabla_{y}}\\ W_{u}^{\nabla_{x}} & W_{v}^{\nabla_{y}} & \epsilon D \end{array}\right]\left[\begin{array}{c} U\\ V\\ P \end{array}\right]=\left[\begin{array}{c} F_{1}\\ F_{2}\\ 0\; \end{array}\right]. \end{array} \end{array}$$
where B is the left-end weight matrix, $W_{u}^{\Delta +\nabla }$ and $W_{v}^{\Delta +\nabla }$ are (N×N) matrices composed of weight coefficients and boundary condition coefficients used to approximate *u* and *v* in Equations (15) and (16), respectively, $W_{u}^{\nabla_{x}}$ and $W_{v}^{\nabla_{y}}$ are a matrix of (N×N) composed of weighted coefficients for approximating $\frac{\partial u}{\partial x}$ and $\frac{\partial v}{\partial y}$, $W_{p}^{\nabla_{x}}$ and $W_{p}^{\nabla_{y}}$ are a matrix of (N×N) composed of weighted coefficients for approximating $\frac{\partial p}{\partial x}$ and $\frac{\partial p}{\partial y}$, $F_{1}$ and $F_{2}$ are known right-end terms, **U** is the solution of velocity *u* on point set X, **V** is the solution of velocity *v* on point set X, **P** is the solution of pressure *p* on point set X, *D* is the unit matrix of (N×N), and ϵ is the stabilization parameter, ϵ = 1.0 × 10${}^{-6}$.

#### 3.2.2. Selection of Degrees of Added Polynomial and Number of Template Nodes

According to [13,14], without depending on the dimension, when using radial basis function PHS: $\Phi \left(r\right)=r^{p}$, when *p* is an odd number and the l-degree polynomials are supplemented to approximate the *k*-order derivative the convergence rate is not determined by the order of PHS, and is instead determined by the highest degree of the polynomial used. The solution accuracy is $\;O(h^{l-k+1})$. For the PDE with only the first-order spatial derivative, the convergence speed can reach $\;O(h^{l})$, while for the PDE with the second-order spatial derivative, the convergence speed can reach $\;O(h^{l-1})$. The number of terms of the l-degree polynomial is $\;\frac{\left(l\;+\;1)(l\;+\;2\right)}{2}$. When approximating the value of an operator acting on the center point of the template, symmetry of the node templates is beneficial, as symmetrical node templates provide uniform information. In this paper, the numerical examples are considered in two layouts, namely, a right-angle node layout and a hexagonal node layout, with the node layouts shown in Figure 1. The right-angled node layout has completely symmetrical templates on numbers 5, 9, 13, 21, 25, 29, 37, 45, etc., while the hexagonal node layout has completely symmetrical templates on numbers 7, 13, 19, 31, 37, 43, 55, 61, etc. In this paper, the minimum distance between two points under the right-angled node layout is $\;h=\frac{1}{\sqrt{N}\;-\;1}$, while *N* is the total number of nodes.

## 4. Numerical Method

In this section, several numerical examples are used to demonstrate the effectiveness of the proposed methods; the relative error $L_{2}$ and error $L_{\infty }$ are applied for comparison of the numerical solution and true solution. The relative error $L_{2}$ is defined as $error_{L_{2}}=\frac{∥u-u_{h}{∥}_{2}}{{∥u∥}_{2}}$, while the error $L_{\infty }$ is defined as $\;error_{L_{\infty }}=∥u-u_{h}{∥}_{L_{\infty }\left(\Omega \right)}=\underset{x\in X}{max}\left|u\left(x\right)-u_{h}\left(x\right)\right|$, where $\;u_{h}$ is the obtained numerical solution and u is the true solution x∈X.

### 4.1. Convergence Test

This example studies the two-dimensional Navier–Strokes equations under the layout of right-angled nodes. We assume that Equation (Equation 1) has the true solutions
(21)$$\begin{array}{c} \begin{array}{c} u=\left(x^{2}-2x^{3}+x^{4}\right)\left(2y-6y^{2}+4y^{3}\right)\;\;\;\mathrm{on}\;\Omega ,\\ v=-\left(y^{2}-2y^{3}+y^{4}\right)\left(2x-6x^{2}+4x^{3}\right)\;\;\;\mathrm{on}\;\Omega ,\\ p=\cos\left(\pi x\right)\sin\left(\pi y\right)\;\;\;\mathrm{on}\;\Omega . \end{array} \end{array}$$
where $\Omega =\left[0,1\right]\times \left[0,1\right]$, the viscosity coefficient ν=1, and the nodes are uniformly distributed on Ω as shown in Figure 1 left. For purpose of the convergence test, the basis function $\;\Phi \left(r\right)=r^{7}$ is applied to add third-order and second-order polynomials for approximation of the velocity u=(u,v), while the basis function $\;\Phi \left(r\right)=r^{5}$ is applied to add second-order and first-order polynomials for approximation of the pressure p. For the number of node templates approximated at a central node of the velocity and pressure in Equations (15) and (16), we select $\;n_{1}=21$ and $\;n_{2}=9$, respectively.

Figure 2 left shows the addition of the third-order polynomial for approximation of the velocity and the error results for pressure approximation when adding the second-order polynomial. The figure shows the relative error $L_{2}$ when the number of nodes *N* is 25, 81, 289, 1089, and 4225. The error decreases with increasing number of nodes, and the convergence order of velocity and pressure reaches third-order and second-order. When the number of nodes *N* is 4225, the velocity error is $4.4583\times 10^{-5}$ and the pressure error is $2.2254\times 10^{-2}$. The right0hand side of Figure 2 shows the error results of the approximation when adding the second-order polynomial to the velocity and adding the first-order polynomial to the pressure. When the number of nodes *N* is 4225, the relative errors $L_{2}$ of the velocity and pressure are $4.2393\times 10^{-5}$ and $2.8000\times 10^{-2}$, respectively.

By comparing the two figures, the error between the numerical solution and the true solution obtained from approximation by adding third-order and second-order polynomials to the velocity and pressure is far less than that obtained from approximation by adding second-order and first-order polynomials to the velocity and pressure. When the number of nodes is less than 1089, the former reaches the corresponding convergence order. As the number of nodes approaches 4225, the order of velocity is less than 2 due to the reduction in point density. This example illustrates that under the right-angled node layout, the radial basis function difference method based on Oseen iteration is convergent and effective for solving incompressible Navier–Stokes equations.

### 4.2. Small Viscosity Problem

In order to test the suitability of the method proposed in this paper for small viscosity problems, we begin by assuming that Equation (Equation 1) has a true solution (21). By setting the viscosity coefficient ν=0.00001, the nodes are distributed in a right-angled node layout. In this example, the basis function $\;\Phi \left(r\right)=r^{7}$ is applied to add the third-order polynomial for approximation of the velocity u=(u,v), and the basis function $\;\Phi \left(r\right)=r^{5}$ is applied to add the second-order polynomial for approximation of the pressure *p*. Table 1 shows the error results when ν is 0.00001 and number of nodes *N* is 25, 81, 289, 1089, and 4225. When the number of nodes *N* is 4225, the relative error $L_{2}$ of the velocity is $6.7100\times 10^{-5}$ and the relative error $L_{2}$ of the pressure is $2.0271\times 10^{-2}$. Figure 3 shows the corresponding error order.

### 4.3. Hexagonal Node Layout

In this section, in order to test the suitability of the method proposed in this paper for different node layouts, we consider the two-dimensional equation under the hexagonal node layout, as shown in Figure 1 right, assuming that Equation (Equation 1) has a true solution (21) where $\Omega =\left[0,1\right]\times \left[0,1\right]$ and the viscosity coefficient ν=1. In this example, the basis function $\;\Phi \left(r\right)=r^{7}$ is applied to add the third-order polynomial for approximation of the velocity u and the basis function $\;\Phi \left(r\right)=r^{5}$ is applied to add the second-order polynomial for approximation of the pressure *p*. For the velocity and pressure in a discrete equation, the number of node templates approximated at a central node ($n_{1}$ and $n_{2}$) are separately selected as $\;n_{1}=31$ and $\;n_{2}=13$. When number of nodes is 4699, the relative error $L_{2}$ of the velocity is $2.8638\times 10^{-5}$ and the relative error $L_{2}$ of the pressure is $1.3900\times 10^{-2}$. Figure 4 shows the relative error $L_{2}$ between the numerical solution and the true solution when the number of nodes N is 68, 279, 1166, and 4699.

In summation, these numerical experiments show that the radial basis function finite difference method based on Oseen iteration has good performance for solving Navier–Strokes equations under the hexagonal node layout.

## 5. Conclusions

This paper proposes a radial basis function finite difference method based on Oseen iteration for solving two-dimensional steady Navier–Stokes equations with discretization of the spatial operators of the Navier–Stokes equations using the radial basis function finite difference method with polynomial, then linearizing the equation based on Oseen iteration. In this paper, we provide numerical solutions for the Navier–Stokes equations under right-angled and hexagonal node layouts, analyze and compare the influence of the number of polynomial additions on the accuracy and convergence of the solution, and verify the effectiveness of the proposed method. Under the two node layouts studied in this paper, the proposed method offers high-precision numerical solutions for solving Navier–Stokes equations while demonstrating good performance. Its extension to unsteady problems will be the focus of our future research work.

## Figures and Tables

**Figure 1 entropy-25-00804-f001:**
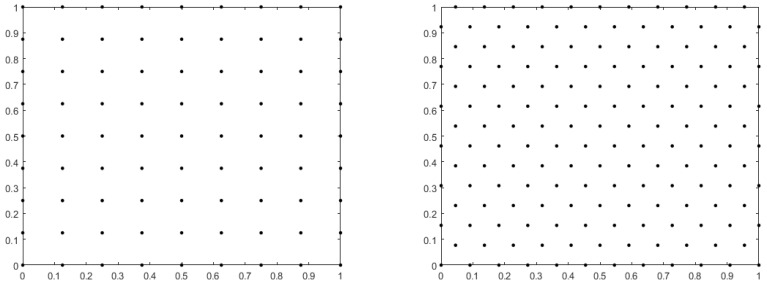
Right-angled node layout (**left**) and hexagonal node layout (**right**).

**Figure 2 entropy-25-00804-f002:**
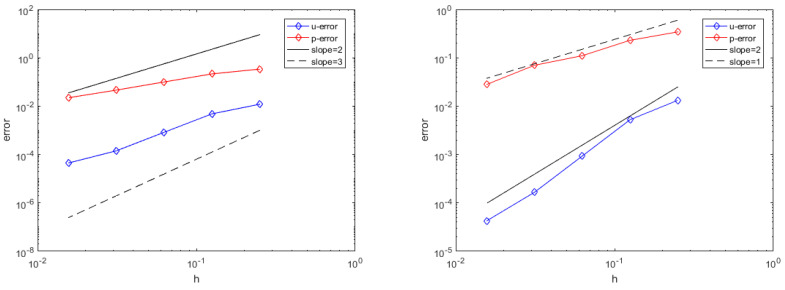
Relative error $L_{2}$ of the velocity *u* and pressure *p* (right-angled node layout).

**Figure 3 entropy-25-00804-f003:**
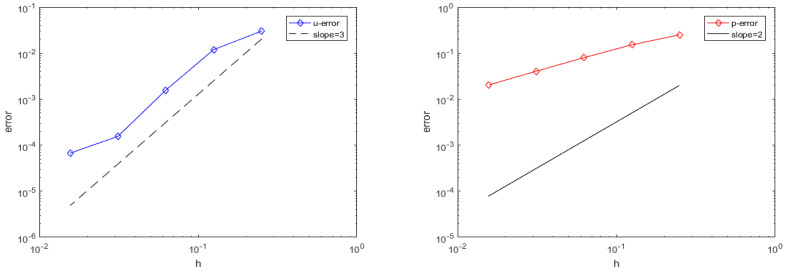
Relative error $L_{2}$ of the velocity u (**left**) and the pressure *p* (**right**) (ν=0.00001).

**Figure 4 entropy-25-00804-f004:**
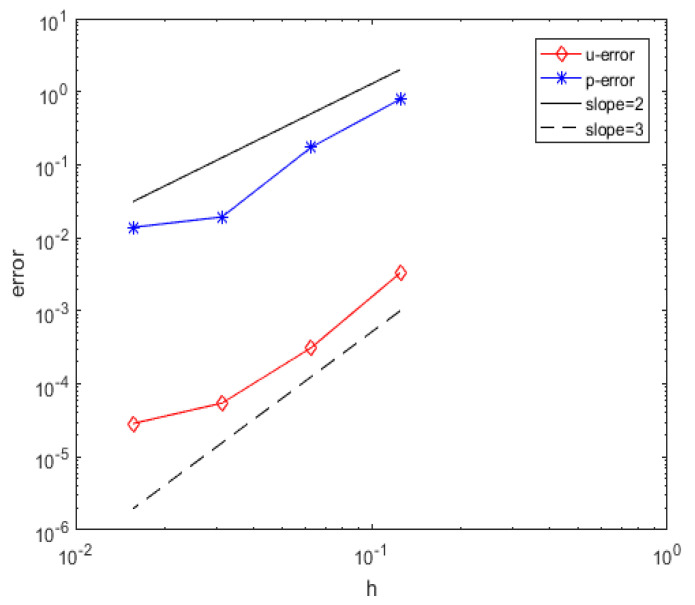
Relative error $L_{2}$ of the velocity u and pressure *p* (hexagonal node layout).

**Table 1 entropy-25-00804-t001:** Relative error $L_{2}$ and error $L_{\infty }$ between numerical solution and true solution of the velocity *u* and pressure *p* (ν=0.00001).

*N*	Relative Error $L_{2}$ of *u*	Relative Error $L_{2}$ of *p*	$L_{\infty }$ Error of *u*	$L_{\infty }$ Error of *p*
25	$2.9942\times 10^{-2}$	$2.4962\times 10^{-1}$	$9.9977\times 10^{-1}$	$1.0191\times 10^{-1}$
81	$1.1883\times 10^{-2}$	$1.5148\times 10^{-1}$	$1.9150\times 10^{-1}$	$3.3872\times 10^{-2}$
289	$1.5680\times 10^{-3}$	$7.9769\times 10^{-2}$	$1.2605\times 10^{-2}$	$9.4008\times 10^{-3}$
1089	$1.5564\times 10^{-4}$	$4.0410\times 10^{-2}$	$6.2549\times 10^{-4}$	$2.4502\times 10^{-3}$
4225	$6.7100\times 10^{-5}$	$2.0271\times 10^{-2}$	$1.3483\times 10^{-4}$	$6.2380\times 10^{-4}$

## Data Availability

Not applicable.

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
