# Peer review of "Radial Basis Function Finite Difference Method Based on Oseen Iteration for Solving Two-Dimensional Navier–Stokes Equations"

_entropy, 2023, doi:10.3390/e25050804_

Round 1

Reviewer 1 Report

The manuscript considers the numerical integration of the Navier-Stokes equations. A complete substantiation of the proposed new method is given. I believe that the manuscript can be published if the authors justify the motives for choosing particular examples to illustrate the numerical method.

Author Response

Thank you. I have revised the manuscript.

Reviewer 2 Report

This paper is concerned with the radial basis function finite difference
method is used to solve two-dimensional steady incompressible Navier-Stokes equations. The authors describe that the radial basis function finite difference method with polynomial and obtain several numerical examples to verify the convergence and effectiveness of the radial basis function finite
difference method based on Oseen Iteration.

I think that the result of this paper is interesting and important
 in the study of the numerical method for the Navier-Stokes equations.

However In this paper, there are some unclear descriptions described below, so I think that they should be added or revised to make it easier for readers to understand. Please see attachment file.

Author Response

Hello, I have finished modifying.

Round 2

Author Response

Hello, I have finished revising the manuscript.
